# LEVIS : Large Exact Verifiable Input Spaces for Neural Networks

**Mohamad Chehade** [* 1 2]  **Wenting Li** [* 2]  **Brian W. Bell** [2]  **Russell Bent** [2]  **Saif R. Kazi** [2]  **Hao Zhu** [1]

## Abstract

The robustness of neural networks is crucial in safety-critical applications, where identifying a reliable input space is essential for effective model selection, robustness evaluation, and the development of reliable control strategies. Most existing robustness verification methods assess the worst-case output under the assumption that the input space is known. However, precisely identifying a verifiable input space $\mathcal{C}$, where no adversarial examples exist, is challenging due to the possible high dimensionality, discontinuity, and non-convex nature of the input space. To address this challenge, we propose a novel framework, LEVIS, consisting of LEVIS$-\alpha$ and LEVIS$-\beta$. LEVIS$-\alpha$ identifies a single, large verifiable ball that intersects at least two boundaries of a bounded region $\mathcal{C}$. In contrast, LEVIS$-\beta$ systematically captures the entirety of the verifiable space by integrating multiple verifiable balls. Our contributions are fourfold: (1) We introduce a verification framework, LEVIS, incorporating two optimization techniques for computing nearest and directional adversarial points based on mixed-integer programming (MIP). (2) To enhance scalability, we integrate complementarity-constrained (CC) optimization with a reduced MIP formulation, achieving up to a 6-fold reduction in runtime while approximating the verifiable region in a principled manner. (3) We provide a theoretical analysis characterizing the properties of the verifiable balls obtained through LEVIS$-\alpha$. (4) We validate our approach across diverse applications, including electrical power flow regression and image classification, demonstrating performance improvements and visualizing the geometric properties of the verifiable region.

## 1. Introduction

Despite the remarkable growth and transformative impact of neural network models, their deployment in safety-critical domains remains limited. An example is neural network-based controllers for electric inverters that have significant potential to support the integration of renewable energy resources (Cui et al., 2022). These applications demand extremely high accuracy and reliability, as even minor errors can lead to millions of dollars in economic losses or large-scale power grid failures. However, neural networks are inherently vulnerable to data variations, including both natural perturbations and adversarial attacks. For instance, slight noise added to input data can lead to significant misclassification errors (Goodfellow et al., 2015). Even state-of-the-art foundation models can achieve less than 80% accuracy when evaluated on perturbed inputs (Liu et al., 2023).

One promising approach to addressing these robustness challenges is *neural network verification*. Most existing research focuses on verifying whether the *outputs* of a neural network satisfy specific worst-case criteria within a predefined input domain (Tjeng et al., 2019; Gowal et al., 2018). Techniques range from transforming nonlinear activations into integer constraints to compute worst-case outputs, to using linear (Wang & et.al., 2021) or quadratic (Kuvshinov & Günnemann, 2022) relaxations to estimate bounds. Advanced methods further integrate optimization techniques like bound tightening and leverage parallelism for speedup (Zhang et al., 2022). However, these approaches typically assume the input domain is predefined and do not explicitly evaluate the structure of the input space itself, potentially leading to under- or over-estimations of robust input regions.

As illustrated in Figure 1, the verifiable input space $\mathcal{C}$ contains all inputs $x \in \mathcal{C}$ that produce outputs satisfying a given specification, such as $f(x) > 0$ (shown by the red boundary). However, $\mathcal{C}$ is often non-convex, discontinuous, and high-dimensional, making it difficult to characterize precisely. Early work explored the input space primarily to identify adversarial vulnerabilities (Peck et al., 2017), while

[*]Equal contribution [1]Chandra Department of Electrical and Computer Engineering, The University of Texas at Austin, Austin, TX, USA [2]Los Alamos National Laboratory, Los Alamos, NM, USA. Correspondence to: Mohamad Chehade <chehade@utexas.edu>, Wenting Li <wenting@lanl.gov>, Brian W. Bell <bwbell@lanl.gov>, Russell Bent <rbent@lanl.gov>, Saif R. Kazi <skazi@lanl.gov>, Hao Zhu <haozhu@utexas.edu>.

*Proceedings of the 42nd International Conference on Machine Learning*, Vancouver, Canada. PMLR 267, 2025. Copyright 2025 by the author(s).

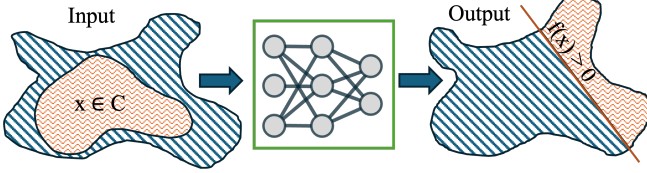

Input      Output

$x \in C$

*Figure 1.* The verifiable input space (orange) is $\mathcal{C}$, consisting of the inputs $x \in \mathcal{C}$ that produce outputs satisfying the condition $f(x) > 0$, determined by the red line. Note: the left orange space may only be a subset of the pre-image of the right orange space.

more recent research has aimed to define verifiable input regions meeting physical or accuracy-based criteria. These include abstraction-based approaches (Matoba & Fleuret, 2020), over-approximation of verifiable regions (Kotha et al., 2024; Sundar et al., 2023), and probabilistic verification techniques (Grunbacher et al., 2021; Bell et al., 2025). Nevertheless, the task of precisely identifying input regions that are entirely free of adversarial points remains largely unexplored.

To address this gap, we propose a novel framework, LEVIS (Large and Exact Verifiable Input Spaces), to identify verifiable input regions for neural networks. LEVIS consists of two algorithms: LEVIS-$\alpha$ and LEVIS-$\beta$. LEVIS-$\alpha$ identifies a single, large verifiable ball that intersects the true space $\mathcal{C}$ at two boundaries, capturing a substantial region guaranteed to be verifiable. In contrast, LEVIS-$\beta$ extends this idea by constructing a collection of verifiable balls to cover the entire input space, which may be high-dimensional, nonconvex, and discontinuous. Importantly, every point in the regions identified by LEVIS is guaranteed to satisfy the given verification criteria.

To this end, we develop a new optimization framework that computes the largest verifiable ball at a given center, locates adversarial points along specific directions, and scales effectively to large-scale problems. Our contributions are summarized as follows:

1. We develop a novel optimization framework to accurately locate the largest verifiable ball centered at a known input $x_0 \in \mathbb{R}^n$, using mixed-integer programming (MIP) with global optimality guarantees. We further incorporate directional constraints into the framework to locate adversarial points along specific directions.

2. To enhance scalability, we propose a complementarity-constrained (CC) relaxation that formulates an approximate verifier. Unlike other approximate methods, our approach explicitly characterizes the conditions under which the solution remains globally optimal. This relaxation achieves up to a 6× speedup while maintaining

a negligible error in global optimality (experimentally within 0.004). The core idea is to reformulate ReLU activations using CC constraints, reducing the number of integer variables in the MIP formulation.

3. We propose two search strategies: LEVIS-$\alpha$, which seeks a single large verifiable ball centered in the interior of $\mathcal{C}$, and LEVIS-$\beta$, which aggregates multiple verifiable balls to cover the full verifiable input space.

4. We evaluate our methods on two neural network applications—electric power flow regression and image classification—demonstrating both performance gains and visual insights into the geometry of the verifiable input space.

## 2. Background and Related Work

**Preliminary Notation.** We use $\mathcal{C}$ and $\mathcal{V}$ to denote the verifiable region and the union of verifiable balls, respectively. $\mathcal{B}(c)$ represents a ball centered at $c$. The vector $e_k$ is a standard basis vector where the $k$-th position is 1 and all others are 0. The $\|x\|_p$ norm is defined as $(\sum_{i=1}^{d} |x_i|^p)^{1/p}$ for $p = 1, 2$, and $\|x\|_\infty = \max_i |x_i|$. We define an $L$-layer rectified linear (ReLU) neural network, where layer $i$ (for $i = 1, \ldots, L$) uses weights $W^i \in \mathbb{R}^{d_i \times d_{i-1}}$ and biases $\beta^i \in \mathbb{R}^{d_i}$. The pre-activation and post-activation outputs for layer $i$ are given by $z^i = W^i \hat{z}^{i-1} + \beta^i$ and $\hat{z}^i = \sigma(z^i) = \max(z^i, 0)$, respectively. The input $\hat{z}^0$ equals the input data $x \in \mathbb{R}^{d_0}$, and the output of the network is $f(x) = z^L \in \mathbb{R}^{d_L}$.

**Bounds on the Output Domain: Verification of Neural Networks.** Bounding the outputs of neural networks (NNs) is an effective strategy for verifying robustness and enhancing model reliability. In standard verification, given an input region $\mathcal{S}$, the output must satisfy a predefined specification $\mathcal{P}$ to ensure safety or correctness. This is typically framed as an optimization problem: find $f^* = \min_{x \in \mathcal{S}} f(x)$, subject to $\hat{z}^0 = x$, $z^i = W^i \hat{z}^{i-1} + \beta^i$, and $\hat{z}^i = \max(z^i, 0)$ for $i = 1, \ldots, L$. The input domain is defined as $\mathcal{S} = \{x \mid \|x - x_0\|_\infty \leq \varepsilon\}$, and the network is considered verified if the condition $\mathcal{P} = \{f^* > 0\}$ holds.

Verification of neural networks is NP-complete, primarily due to nonlinear activation functions. Solution approaches can be categorized into three types: exact or complete verifiers (Tjeng et al., 2019), approximate or incomplete verifiers (Gowal et al., 2018; Wang & et.al., 2021), and probabilistic verifiers (Grunbacher et al., 2021; Marzari et al., 2024). Unlike these methods, which focus on confirming output correctness for a known input domain, our work prioritizes input domain analysis to identify the largest possible region that yields verifiable outputs, ensuring all points within this region meet the satisfaction criteria.

**Bounds on the Input Domain: Robustness to Adversarial Perturbations.** Inputs that, when perturbed, result in erroneous neural network outputs are termed *adversarial examples*. The subset of inputs free from adversarial examples forms the *verifiable input region*, which is crucial for robust model selection and training (Kuvshinov & Günnemann, 2022; Li et al., 2023). Early works primarily focused on measuring the maximum perturbation magnitudes that induce adversarial examples rather than characterizing the global structure of the verifiable input region (Kuvshinov & Günnemann, 2022). A recent approach approximates the input space using a convex hull derived via bound propagation techniques (Kotha et al., 2024). However, this method results in an *over-approximation* of the verifiable region, meaning that some points within the convex hull may not actually be verifiable.

## 3. Problem Formulation

A *verifiable space* comprises input data points for neural networks that consistently yield verifiable outputs. Precisely identifying large verifiable spaces is crucial for model selection, robustness evaluation, and safe control operations (Kotha et al., 2024). The challenge lies in maximizing the verifiable input space $\mathcal{C}$, where all data points in $\mathcal{C}$ generate outputs that satisfy verification guarantees. This objective can be formalized as $\max_{\mathcal{C}} \min_{x \in \mathcal{C}} f(x) > 0$, which defines a fundamentally intractable min-max optimization problem, especially when $\mathcal{C}$ is non-convex.

To manage this complexity, we propose approximating the verifiable space using one or more verifiable balls. Each ball is defined by a center and radius such that all data points within it produce verifiable outputs when passed through the neural network. Crucial to our approach is the identification of centers and radii that ensure (1) the verifiability of all interior points under specified conditions, and (2) scalability to high-dimensional, non-convex input regions.

## 4. Proposed Approach

Our method begins by precisely identifying a maximal verifiable ball centered at a given point $x_0$, denoted as the center $c$, and then iteratively shifts this center to either identify a large boundary-touching verifiable ball (LEVIS-$\alpha$) or assemble a large union of such verifiable balls (LEVIS-$\beta$).

### 4.1. Find a Maximum Verifiable Input Ball around c

We find the maximum verifiable ball around center $c$ by locating the **nearest adversarial data point** $x^*$. This point lies on the boundary of the verifiable input space and yields a violating output $f(x^*) \leq 0$. The optimization problem for

finding $x^*$ is structured as follows:

$$\min_x \quad \|x - c\|_p \tag{1a}$$

$$\text{subject to} \quad \hat{z}^0 = x \tag{1b}$$

$$z^i = W^i \hat{z}^{(i-1)} + \beta^i, \quad i = 1, \ldots, L \tag{1c}$$

$$\hat{z}^{(i)} = \sigma(z^{(i)}), \quad i = 1, \ldots, L-1 \tag{1d}$$

$$f(x) = z^L \tag{1e}$$

$$f(x) \leq 0 \tag{1f}$$

Equation (1f) ensures the solution violates the specified output condition, i.e., $x$ is adversarial. The objective in (1a) minimizes the $l_p$-norm distance from the center $c$, seeking the nearest such adversarial point.

To solve this nonlinear program, we leverage a mixed-integer programming (MIP) formulation by explicitly encoding the ReLU activation functions. ReLU is a piecewise linear function: for $\hat{z} = \text{ReLU}(Wz + b)$, we have $\hat{z} = Wz + b$ if $Wz + b \geq 0$, and $\hat{z} = 0$ otherwise. This behavior can be encoded in a MIP by introducing a binary variable $a$, where $a = 1$ if $Wz + b \geq 0$ and $a = 0$ otherwise. Using the Big-M method, this conditional logic is translated into a set of linear constraints, thereby enabling the global optimization of ReLU networks using standard MILP solvers (Tjeng et al., 2019). Similar formulations have been used for proving equivalence between networks (Kleine Büning et al., 2020) and for finding the closest input that yields a target label (Szegedy et al., 2014).

The verifiable ball, denoted as $\mathcal{B}(c)$, is characterized by the center $c$ and radius $r = \|x^* - c\|_p$. All points inside $\mathcal{B}(c)$ are guaranteed to produce verifiable outputs, with $x^*$ representing the closest violation. To improve the scalability of the MIP solver, we propose a hybrid approach that integrates efficient nonlinear programming (NLP) with a reduced MIP formulation, achieving both computational efficiency and guaranteed optimality.

### 4.2. Fast Solver with complementarity Constraints

We represent the ReLU function using complementarity constraints (CC): $0 \leq \hat{z}^i \perp \hat{z}^i - z^i \geq 0$, where the "$\perp$" operator enforces that at least one of the two inequalities holds as an equality; that is, $\hat{z}^i = 0$, $\hat{z}^i = z^i$, or both. This CC-based formulation replaces integer variables with continuous functions, enabling a more scalable neural network representation (Yang et al., 2022; Kilwein et al., 2023). Although standard nonlinear programming (NLP) solvers can handle this formulation, they may converge to spurious stationary points with feasible descent directions (Leyffer & Munson, 2007). To mitigate this, we adopt the two-step hybrid strategy (Kazi et al., 2024), which integrates the NLP formulation with a reduced mixed-integer programming (MIP) approach. This strategy ensures global optimality

under certain conditions while improving computational efficiency.

Specifically, let $p^i$ and $q^i$ be the complementarity variables for the $i$th neuron. The nonlinear optimization of (1) transforms into:

$$\min_{x, p^i, q^i} \quad \text{(1a)}$$

$$\text{s.t.} \quad z^i = p^i - q^i, \quad \hat{z}^i = p^i \quad \text{(2a)}$$

$$p^i q^i \leq 0, \quad p^i, q^i \geq 0 \quad \text{(2b)}$$

$$\text{(1b), (1c)–(1f)} \quad \text{(2c)}$$

where constraints in (2b) represent a nonlinear equivalent of the ReLU function. To improve convergence behavior, we introduce a small regularization parameter $\varepsilon > 0$ (suggested to be $10^{-5}$), modifying the constraint to $p^i q^i \leq \varepsilon$ (Scholtes, 2001). This reduces the problem complexity from an NP-hard integer formulation to polynomial scaling in the number of neurons, $N$.

However, NLP solvers may still converge to locally optimal solutions and spurious points where the bi-active set is non-empty, i.e., $I_0 \equiv \{j \mid p^i_j = q^i_j = 0\}$. To address this, we classify neurons into three groups: if $p^i_j > 0$ and $q^i_j = 0$, then $j \in I^i_+$; if $p^i_j = 0$ and $q^i_j > 0$, then $j \in I^i_-$; and if $p^i_j = q^i_j = 0$, then $j \in I^i_0$. We then construct a reduced MIP that introduces integer variables only for neurons in the set $I_0$, balancing efficiency and optimality. This reduces the number of integer variables from $N$ to $|I_0|$, which is typically a small subset, as confirmed in Section 5.2. We assert that the solution is globally optimal and verifiable for all inputs within the ball, provided the neuron activation states—active (in $I_+$) or inactive (in $I_-$)—remain consistent with the sets $I_+$ and $I_-$.

In terms of computational complexity, the NLP formulation derived from neural network verification exhibits substantial sparsity in its constraint Jacobian, owing to the localized connectivity and layered architecture of neural networks. As both the number of decision variables and constraints scale linearly with the total number of neurons, denoted by $N$, we characterize the overall complexity in terms of $N$. When solved using a sparse interior-point method such as IPOPT (Wächter & Biegler, 2006), the computational complexity is primarily governed by the cost of factorizing the Karush–Kuhn–Tucker (KKT) system at each iteration. This cost depends critically on the sparsity pattern and numerical properties of the constraint matrix. In favorable cases—such as when the matrix has a particular structure that permits efficient factorization—the per-iteration cost can be reduced to $O(N \log N)$ using solvers like MA57 (Duff, 2004). In contrast, for less structured or denser matrices, the cost may grow toward $O(N^2)$.

## 4.3. LEVIS: Large Exact Verifiable Input Spaces for Neural Networks

LEVIS employs two search strategies for identifying verifiable balls. The key innovation lies in dynamically updating the centers based on adversarial points along different directions, ensuring the verifiability of the newly selected centers. We first present the optimization formulation for computing directional adversarial points, followed by a detailed discussion on how these points guide the search trajectory across various geometric structures of the input space.

### 4.3.1. DIRECTIONAL ADVERSARIAL POINT

Our LEVIS search algorithms are guided by adversarial points along specific directions, referred to as *directional adversarial points*. To achieve this, we incorporate directional constraints into the original optimization problem in (1), which dictate the search direction. Given the center of the previous verifiable ball $c$ and an adversarial point $b$, the next adversarial point $\hat{b} \in \mathbb{R}^n$ is determined along a direction that forms a specified angle $\theta$ with the vector $\overrightarrow{bc}$.

Precisely, we construct a general directional constraint and formulate the optimization for the *directional adversarial point* $\hat{b}$ in (3), where the search direction forms a user-defined angle $\theta$ with $\overrightarrow{bc}$. To define this direction, we introduce two orthogonal vectors $d, q \in \mathbb{R}^n$: the vector $d$, aligned with $\overrightarrow{bc}$, is given by $d = \frac{c-b}{\|c-b\|_\infty}$. The vector $q$ represents an orthogonal direction. It can be explicitly specified or generated by sampling a random vector $\xi \sim \mathcal{N}(0, 1)$ and removing its component along $d$, yielding $q = \xi - \frac{\xi^T d}{d^T d} d$, where the second term represents the projection of $\xi$ onto $d$.

Using these definitions, any direction can be expressed as $\phi(\theta) = d \cos(\theta) + q \sin(\theta)$, and the directional constraint for $\hat{b}$ is given by $\hat{b} = c + k\phi$. Special cases of this formulation include: when $\theta = 0$, the constraint reduces to the collinear direction; when $\theta = 90°$, it corresponds to the orthogonal constraint, i.e., $(\hat{b}-c) \cdot (b-c) = 0$. This formulation enables flexible directional adversarial search while maintaining mathematical rigor and generalizability.

$$\min_{\hat{b}, k > 0} \quad \|\hat{b} - c\|_p \quad \text{(3a)}$$

$$\text{subject to} \quad \text{(1b)} - \text{(1f)}, \quad x = \hat{b} \quad \text{(3b)}$$

$$\hat{b} = c + k\phi(\theta) \quad \text{(3c)}$$

$$\phi(\theta) = d \cos(\theta) + q \sin(\theta) \quad \text{(3d)}$$

$$d = \frac{c-b}{\|c-b\|_\infty}, \quad q = \xi - \frac{\xi^T d}{d^T d} d \quad \text{(3e)}$$

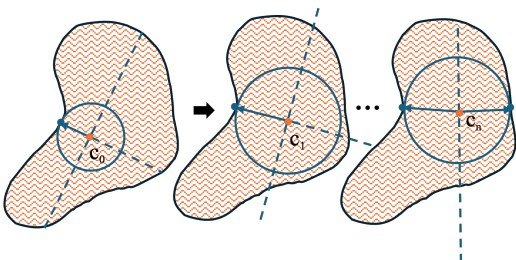

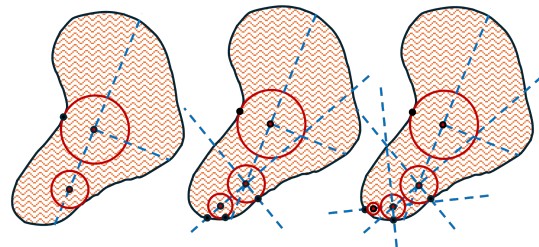

*Figure 2.* Illustration of the sequence of balls centered at the red points $c_0, \cdots, c_n$ obtained by LEVIS$-\alpha$, where the arrows point to the nearest adversarial points that define the radius of those balls. The search converges at the final ball $\mathcal{B}(c_n)$, which touches at least two boundaries of the bounded verifiable input space.

### 4.3.2. (1) LEVIS$-\alpha$: SEARCH FOR THE LARGE BOUNDARY-TOUCHING VERIFIABLE BALL

The core technique of this search strategy is to average $d$ pairs of boundary adversarial points $b_{2j+1}$ and $b_{2j+2}$, $j = 0, \ldots, d-1$, obtained by solving the minimization in (1) with directional constraints. The detailed algorithm, termed LEVIS$-\alpha$, is described in Algorithm 1 and illustrated in Figure 2.

---

**Algorithm 1** LEVIS$-\alpha$: Iterative Refinement for Center Estimation

1: **Initialize:** Given the original data $x_0$ as the ball center $c = x_0$, set $r = \infty$, $r_{old} = 0$, tolerance $\epsilon > 0$, and neural network parameters $\Theta = \{W^i, \beta^i, i = 1, \ldots, L\}$.

2: **while** $\|r - r_{old}\| \geq \epsilon$ **do**
3: $\quad b^1 \leftarrow$ Solve (1), set $r = \|b^1 - c\|_p$
4: $\quad b^2 \leftarrow$ Solve (3) given $b^1$ and $\theta = 90°$
5: $\quad$ **for** $j = 1$ to $d - 1$ **do**
6: $\quad\quad b_{2j+1} \leftarrow$ Solve (3) given $b_{2j-1}$, $\theta = 90°$
7: $\quad\quad b_{2j+2} \leftarrow$ Solve (3) given $b_{2j}$, $\theta = 0°$
8: $\quad$ **end for**
9: $\quad c \leftarrow \frac{1}{2d} \sum_{l=1}^{2d} b_l$, $r_{old} \leftarrow r$
10: **end while**
11: **Return** $c, r$

---

Algorithm 1 updates the center of the verifiable ball in each iteration and eventually converges to a large exact verifiable ball in the central region of the verifiable space, touching at least two adversarial points, as illustrated in Figure 2. In each iteration, the algorithm searches for $d$ pairs of adversarial points along collinear and orthogonal directions. The final region obtained from LEVIS$-\alpha$ is rigorously described below.

**Theorem 4.1.** *For any bounded verifiable region $\mathcal{C} \subset \mathbb{R}^d$, the sequence of balls $\{\mathcal{B}(c_n)\}$ generated by LEVIS$-\alpha$ converges to a ball $\mathcal{B}(c)$ that intersects the boundary of $\mathcal{C}$ at least at two points. Specifically, for any $\epsilon > 0$, there exists*

*Figure 4.* LEVIS$-\beta$ searches for the union of verifiable balls guided by directional adversarial points.

*$N$ such that for all $n > N$, $\mathcal{B}(c)$ contains two points almost symmetric about the center $c$, with less than $\epsilon$ mismatch, and these points lie on the boundary of $\mathcal{C}$.*

The rationale is that LEVIS$-\alpha$ progressively averages the collinear adversarial pairs $(b_{2j+1}, b_{2j+2})$ using $c = \frac{1}{2d} \sum_{l=1}^{2d} b_l$, minimizing asymmetry relative to the center in each iteration. Given that $\mathcal{C}$ is bounded, the search halts when at least one pair $(b'_1, b'_2)$ touches the surface of the ball, while other pairs become symmetric with less than $\epsilon$ mismatch. Hence, the final ball intersects the boundary of $\mathcal{C}$ at at least two distinct points. The complete proof is provided in Appendix A.

LEVIS$-\alpha$ is suitable for exploring small, bounded verifiable input spaces. In contrast, we introduce LEVIS$-\beta$, a more comprehensive and parallel-friendly strategy that systematically identifies verifiable regions even when they are high-dimensional, non-convex, or discontinuous.

### 4.3.3. (2) LEVIS$-\beta$: COLLECT THE LARGE UNION OF VERIFIABLE BALLS

The innovative idea of this search strategy is to look for a verifiable new center outside the known verifiable union by extensively moving in all directions.

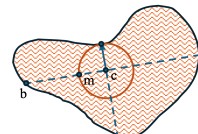

*Figure 3.* Illustration of the geometric position of the surface point.

Specifically, by solving (1) and (3), we obtain a verifiable ball $\mathcal{B}(c)$ centered at $c$ and its nearest directional adversarial point $b$. Once a new ball $\mathcal{B}(c)$ is found, we define the next center to be the point that lies on the same line as $c$ and $b$ but on the internal surface of $\mathcal{B}(c)$, i.e.,

$$m = c - \gamma \frac{c - b}{\|c - b\|_p} \cdot r, \quad \gamma < 1,$$

as illustrated in Figure 3.

Our new Algorithm 2, termed LEVIS$-\beta$, preserves the ver-

---

**Algorithm 2** `LEVIS-β`

---

1: **Initialize:** Given one data point $x_0$ and label $y_0$, the input bounds $[l, u]$, tolerance $\epsilon > 0$, shift factor $\gamma = 0.99$, trained neural network parameters $\Theta = \{W^i, \beta^i, i = 1, \ldots, L\}$, step size $\Delta$, angle $\theta$, and a random seed.

2: Initialize the set of verifiable balls $\mathcal{V} = \emptyset$, let $c = x_0 + \Delta$, and initialize the queue $Q = \{c\}$

3: **while** $Q$ is not empty **do**

4:     Pop the first element of $Q$ as $c$; solve (1) to obtain the radius $r$ centered at $c$ with a corresponding adversarial point $b$.

5:     **if** $r < \epsilon$ **then**

6:         Sample a point from either the range $[l, c]$ or $[c, u]$ and iteratively resample until it is verifiable.

7:         Add this point to $Q$.

8:     **else**

9:         Update the set $\mathcal{V} = \mathcal{V} \cup \{(c, r)\}$.

10:         Solve (3) along the direction $\phi(\theta)$ to find the directional adversarial point $\hat{b}$.

11:         Compute the surface point: $m = c - \gamma \frac{c-b}{\|c-b\|_p} \cdot r$.

12:         Terminate the search if $m \notin [l, u]$, otherwise add $m$ to $Q$.

13:         **while** $m \in \mathcal{V}$ and $\|m - \hat{b}\|_p > \epsilon$ **do**

14:             $m = (m + \hat{b})/2$

15:         **end while**

16:     **end if**

17: **end while**

18: **Return** $\mathcal{V}$

---

ifiability of the center while exploring multiple directions to identify external balls beyond the known union. Step 6 prevents the search from getting stuck at discontinuous or local boundaries before reaching the full extent of the input space, effectively addressing the discontinuity challenge in $\mathcal{C}$. Steps 10–12 update the center using the surface point in the $\phi(\theta)$ direction, ensuring both feasibility (i.e., $m \in [l, u]$) and that $m$ lies within a verifiable ball. Steps 13–15 refine $m$ by nudging it toward $\hat{b}$ if it overlaps with the existing union.

Figure 4 illustrates the expansion of the union over iterations, where blue lines indicate search directions and red circles represent verifiable balls.

**Remark:** We assert that the new center $c$ remains a verifiable point throughout the execution of Algorithm 2. Since $\hat{b}$ is the nearest adversarial point along the direction $\phi(\theta)$, any point closer to the original center $c$ along this direction remains verifiable. The surface point computed in Steps 10–12 satisfies this condition. Moreover, because each new center $c$ is located outside the existing union of verifiable balls, the union's total volume expands progressively during the search. Notably, `LEVIS-β` can be executed in parallel across multiple directions with varying angles $\theta$, shift pa-

rameters $\Delta$, and random seeds to efficiently discover the full verifiable input space, as discussed in Section 5.7.

# 5. Experiments

We implement our algorithms on two benchmark systems designed to meet specific requirements for physical constraints and robust accuracy. We employ three-layer dense neural networks to model solutions for both optimal power flow (regression) and image classification tasks.

For the optimal power flow (OPF) problem, we generate a direct-current (DC) power flow dataset, denoted as $\mathcal{D}_1 = \{(x_k, y_k) \mid x_k \in \mathbb{R}^3, y_k \in \mathbb{R}^3\}_{k=1}^{1000}$, using the IEEE 9-bus power grid benchmark—a widely recognized model for simulating energy flow in power systems. The specification for DC-OPF requires the network output to remain within a predefined limit $y_{\max}$, i.e., $f(x_k) \leq y_k^{\max}$.

For image classification, we utilize the MNIST (LeCun et al., 2010) and CIFAR-10 (Krizhevsky et al., 2009) datasets. MNIST consists of 60,000 training and 10,000 testing samples with an input dimension of 784, while CIFAR-10 includes 50,000 training and 10,000 testing samples with an input dimension of 3,072. The robustness requirement mandates that the predicted probability of the ground-truth class $j$ must be higher than that of an adversarial class $l$, formulated as: $f(x_k)_j - f(x_k)_l > 0$. Further details on the datasets are provided in Appendix B.

To solve the mixed-integer programming (MIP) problems in (1) and (3), we use Gurobi with OMLT (Ceccon et al., 2022) for small neural networks (fewer than 100 neurons) trained on DC-OPF datasets. For large-scale MNIST and CIFAR-10 networks, we adopt the Complementary Constraints (CC) approach (Section 4.2), employing Ipopt (Wächter & Biegler, 2006) for the NLP formulation and Gurobi for the reduced MIP. All optimization models are implemented using Pyomo (Hart et al., 2024).

## 5.1. Visualization of the Exact Verifiable Ball

Figure 5 illustrates the verifiable balls for different $\ell_p$ norms ($p = 1, 2, \infty$) using the DC-OPF dataset in 3D. The analysis provides key insights into selecting $p$ and refining the search strategy. Notably, the volumes of these balls vary significantly, with the $\ell_\infty$ norm producing the largest. We choose $p = \infty$ because its cube-shaped geometry minimizes gaps when adjacent cubes intersect, enhancing coverage. Although our solution to (1) guarantees verifiability within the ball, points equidistant from $x_0$ to $x^*$ may still be adversarial, potentially appearing anywhere on the surface. This highlights the challenge of expanding verifiable regions via vertex enumeration. Instead, our method explores optimal directions to identify new verifiable balls, improving both coverage and efficiency.

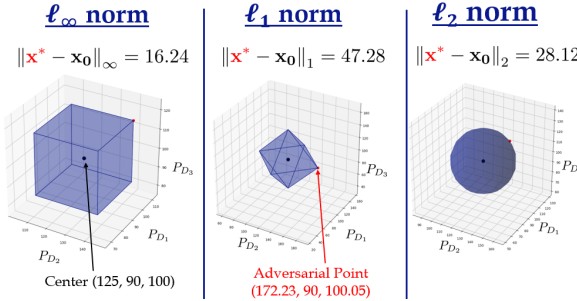

*Figure 5.* Single maximum verifiable balls obtained by solving (1) with different $\ell_p$ norms using the DC-OPF dataset. The center is $x_0 = [125, 90, 100]$, with the nearest adversarial points $x^*$ marked as red nodes. The radius, given by $\|x^* - x_0\|_p$, represents the smallest distance between $x_0$ and $x^*$. In the 3D input space, the verifiable regions for $\ell_\infty$, $\ell_1$, and $\ell_2$ norms (left to right) correspond to a cube, an octahedron, and a sphere, respectively.

## 5.2. Efficiency Comparison

We evaluate the runtime of solving the optimization problem in (1) and compare our method against a strengthened MIP baseline that uses bounds from the CROWN method. Experiments are conducted on image recognition datasets (MNIST and CIFAR-10) using a three-layer ReLU network with 894 neurons.

*Table 1.* Runtime (seconds) and relative speedup (in parentheses) for each method. Speedup is relative to the MIP_CROWN baseline.

| Dataset | MIP$_{\text{CROWN}}$ (s) | Proposed (s) | Optimality Gap |
| --- | --- | --- | --- |
| MNIST | 10.48 | **1.34** (7.82×) | 0.004 |
| CIFAR-10 | 17.47 | **5.96** (2.93×) | 0.0009 |

As shown in Table 1, the proposed method significantly improves efficiency compared to the MIP$_{\text{CROWN}}$ baseline, achieving a 7.82× speedup on MNIST and a 2.93× speedup on CIFAR-10. Our solver (Section 4.2) solves each instance in 1.34 seconds on MNIST and 5.96 seconds on CIFAR-10, while maintaining a small optimality gap. The total runtime includes both a nonlinear programming (NLP) phase and a reduced MIP phase.

## 5.3. Comparison of Radius Size with the Baseline

We compare the size of the verifiable ball obtained by our method with a baseline method based on Lipschitz constants (Fazlyab et al., 2021). As neural networks are Lipschitz continuous with constant $L = \prod_{i=1}^{L} \|W^i\|_p$, we can estimate a lower bound on the adversarial radius as

$$\|x^* - c\|_p \geq \frac{\delta}{L}, \quad \text{where} \quad \delta = \min_{i \neq k} \frac{1}{\sqrt{2}} |(e_k - e_i)^T f(x^*)|.$$

We train neural networks to regress optimal power flow solutions and compare the verifiable radius from our method (solving the NLP in (1)–(1f)) against the Lipschitz-based bound.

Due to the inherent randomness in training neural networks, we repeat the experiment across five independently trained models and report the resulting range of radius values. This repetition ensures robustness of the comparison across different training instances.

Table 2 shows that our method consistently finds verifiable balls with radii up to 10x larger than those estimated by the baseline, underscoring the benefit of solving the exact optimization problem directly.

*Table 2.* Comparison of radii from the exact solver vs. Lipschitz-based lower bound for the DC-OPF network.

| Method | $p = \infty$ | $p = 1$ | $p = 2$ |
| --- | --- | --- | --- |
| Baseline $r$ | 4.52 | 4.80 | 12.88 |
| Our $r$ | **16.24** | **47.28** | **28.12** |

## 5.4. Performance of LEVIS$-\alpha$ for DC-OPF

We implement LEVIS$-\alpha$ for the DC-OPF task and track the variations in the radius of verifiable balls, as shown in Figure 6. Each solid blue line denotes the mean radius across five random implementations of LEVIS$-\alpha$, and the shaded regions represent standard deviations.

Similar behavior is observed for other norms ($p = 1, 2$). To demonstrate the substantial size of the final verifiable ball obtained by LEVIS$-\alpha$, we compare it against two alternative methods: the Exact Fixed Center (EFC) method from (1), and a version of the Lipschitz-based Lower Bound (LLB) method (Fazlyab et al., 2021). Two key observations

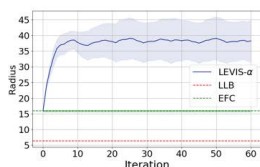

*Figure 6.* Radius over iterations for $\ell_\infty$

emerge: (1) LEVIS$-\alpha$ yields radii 3–9× larger than the baselines, justifying its iterative center refinement in Algorithm 1; and (2) the radius converges, suggesting that the final ball touches two distinct edges of the verifiable region.

## 5.5. Performance of LEVIS$-\beta$ for DC-OPF

We evaluate the statistical behavior of LEVIS$-\beta$ on the DC-OPF task. Across iterations, LEVIS$-\beta$ discovers verifiable balls of varying radii, with smaller balls appearing farther from the original center and closer to the boundary of the verifiable region. Figure 7(a) shows a histogram

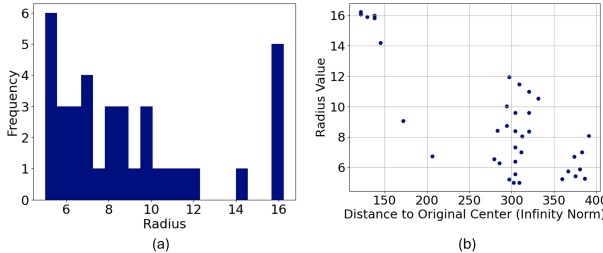

*Figure 7.* Radii distribution for LEVIS-$\beta$: (a) histogram of radii; (b) radii vs. distance to initial center. Smaller-radius balls tend to lie near the boundary of the verifiable region.

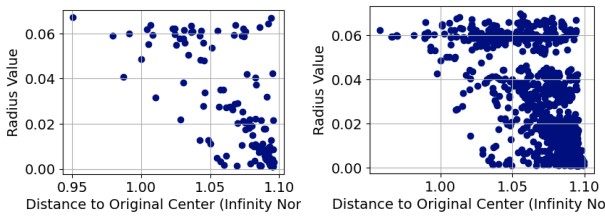

*Figure 8.* Radii distribution from LEVIS-$\beta$ for different directions $\theta \in \{0, 90, 180, 270, 360°\}$ (left) and initialization values $\Delta \in \{0.02, 0.04, 0.06, 0.08\}$ with 10 random seeds per $\Delta$ (right).

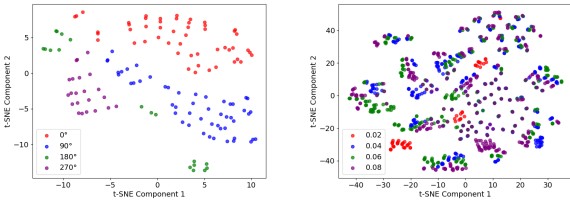

*Figure 9.* t-SNE projection of verifiable ball centers under different search directions $\theta = \{0, 90, 180, 270\}°$ (left) and initialization shifts $\Delta = \{0, 0.02, 0.04, 0.06, 0.08\}$ (right), with 10 runs per setting.

### 5.7. Geometric Characteristics of the Verifiable Input Space

Unlike over- or under-approximation methods, each verifiable ball in LEVIS-$\beta$ is exactly verified and adversarial-free. This allows us to recover structural insights into the geometry of the input space. Figure 9 shows t-SNE projections of verifiable centers based on varying $\theta$ (left) and $\Delta$ (right). The four clusters in Figure 9(left) reflect the orthogonal directions used in the search and suggest regularity in the structure of the verifiable region. In contrast, Figure 9(right) shows that changing $\Delta$ leads to denser but distinguishable center clusters. Together, these plots highlight the structural diversity and robustness of our method in exploring the verifiable space.

## 6. Conclusion and Future Work

Identifying verifiable input regions free from adversarial examples is crucial for model selection, robustness evaluation, and reliable control strategies. This work introduces two MIP-enabled search algorithms, LEVIS$-\alpha$ and LEVIS$-\beta$, to identify verifiable input spaces. We formulate two optimization problems: one for computing the maximum verifiable ball and another for locating nearest adversarial points along specified directions. These guide the LEVIS algorithms in dynamically updating ball centers, enabling efficient exploration of the verifiable input space. We improve MIP scalability by up to 6× via a hybrid NLP-MIP solver, with a worst-case optimality gap below 0.004. Unlike local optimizers, our method ensures global optimality when neuron activation patterns match those from the NLP solution. We visualize verifiable balls under various norms to inform search design and benchmark our method against state-of-the-art techniques in terms of efficiency and radius accuracy. Finally, we analyze the distribution of verifiable balls collected by LEVIS-$\beta$ across scenarios. Future work will adapt LEVIS to neural network-based control systems to certify safe input regions ensuring stability and reliability.

of radii, while (b) plots each radius against its distance to the original center $x_0$. Together, these visualizations indicate that smaller-radius balls dominate the union and are concentrated near the boundaries—highlighting the natural shrinkage of verifiable regions in such areas.

### 5.6. Distribution of Radii Discovered by **LEVIS−$\beta$** in High-Dimensional Space

We analyze the distribution of verifiable ball radii discovered by LEVIS$-\beta$ on the MNIST dataset. Compared to DC-OPF, the radii are generally smaller, which we attribute to the higher input dimensionality and network complexity. We visualize the relationship between radii and distance to the initial center under varying search angles $\theta$, initialization shifts $\Delta$, and random seeds.

Figure 8 shows that most verifiable balls have radii below 0.07. Varying $\theta$, $\Delta$, and random seeds leads to diverse verifiable balls distributed across the input space. Randomness mainly affects the orthogonal vector $q$ in (3d), thereby influencing search directions. Each run is independent and can be parallelized over different $\theta$ and $\Delta$ values. Multi-directional searches can further boost coverage efficiency, which we leave for future work.

## Acknowledgements

The work was funded by Los Alamos National Laboratory's Directed Research and Development project, "Artificial Intelligence for Mission (ArtIMis)'" under U.S. DOE Contract No. DE-AC52-06NA25396, and by NSF Grant 2130706 and ARO Grant W911NF2310266.

## Impact Statement

This work introduces LEVIS, a novel MIP-enabled search framework for identifying verifiable input spaces in neural networks, ensuring robustness against adversarial attacks. By formulating two complementary optimization problems, we efficiently compute the maximum verifiable region and locate directional adversarial points, enabling scalable and reliable model verification. Our approach significantly enhances neural network safety and interpretability, with potential applications in autonomous systems, power grid stability, and secure AI-driven decision-making.

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

# A. Appendix: Theoretical Results

**Theorem A.1.** *For any convex bounded verifiable region $\mathcal{C} \subset \mathbb{R}^d$, the sequence of balls $\{B(c_n)\}$ generated by LEVIS-$\alpha$ converges to a ball $\mathcal{B}(c)$ that intersects the boundary of $\mathcal{C}$ at least at two points. Specifically, for any $\epsilon > 0$, there exists a number of algorithm iterations $N$ such that for all $n > N$, $\mathcal{B}(c)$ encompasses two points almost symmetric about the center $c$ with less than $\epsilon$ mismatch, and these two points lie on the boundary of $\mathcal{C}$.*

*Proof.* Consider the convex, bounded, and closed region $\mathcal{C} \subset \mathbb{R}^d$. At each step $n$ of Algorithm 1, a ball $B(c_n)$ with the radius $r_n$ is generated, where $c_n$ is the center, with the property that $B(c_n) \subseteq \mathcal{C}$. By convexity and boundedness, the boundary $\partial \mathcal{C}$ is non-empty, compact, and ensures that the sequence $\{B(c_n)\}$ must converge to a configuration where the ball touches the boundary of $\mathcal{C}$ at multiple points.

Next, we introduce a constant $\lambda$, which we define as the greatest rate of change in the width of the region $\mathcal{C}$ along any line passing through its center. Specifically, $\lambda$ characterizes the maximum rate at which the distance between two points on the boundary of $\mathcal{C}$, on opposite sides of a line passing through the center, changes as we move the line across $\mathcal{C}$. This constant $\lambda$ is finite due to the convexity and boundedness of $\mathcal{C}$.

Now, consider the rays emanating from the center $c_n$ in the directions aligned with the axes of the $\ell_1$ norm. The algorithm iteratively adjusts the center $c_n$ to reduce the asymmetry of the distances from $c_n$ to the boundary $\partial \mathcal{C}$ along these rays. Let $\Delta_n$ denote the measure of asymmetry at the $n$-th step, defined as the maximum difference between the distances from $c_n$ to $\partial \mathcal{C}$ along opposite rays.

By the convexity of $\mathcal{C}$, moving the center $c_n$ results in a decrease in $\Delta_n$ at a rate proportional to the radius $r_n$ from $c_n$ to the boundary $\partial \mathcal{C}$. Specifically, due to the convexity and the definition of $\lambda$, the reduction in $\Delta_n$ between steps $n$ and $n + 1$ can be bounded below by $\lambda \cdot r_n$. Hence, the sequence $\{\Delta_n\}$ converges to zero as $n$ increases.

To formalize this, given any $\epsilon > 0$, choose $N$ such that for all $n > N$, the asymmetry $\Delta_n$ is less than $\epsilon$. The constant $\lambda$ ensures that for $n > N$, the ball $B(c_n)$ touches the boundary $\partial \mathcal{C}$ at least at two distinct points, as the rays from the center are symmetrized to within $\epsilon/\lambda$. Thus, $B(c_n)$ intersects $\partial \mathcal{C}$ at two points, say $x_n$ and $y_n$, satisfying $\|x_n - y_n\|_1 \geq \frac{r_n}{2}$.

Moreover, since the asymmetry $\Delta_n$ is controlled by $\lambda$, the point $b_n$ within $B(c_n)$ can be chosen such that $\|x_n - b_n\|_1 > \frac{r_n}{2}$ and $\min_{r \in R} \|b_n - r\| < \epsilon$.

Therefore, the sequence $\{B(c_n)\}$ converges to a ball that touches the boundary of $\mathcal{C}$ at at least two distinct points, which completes the proof. $\square$

**Note:** In general, for non-convex regions, this does not necessarily hold. In particular, separate convex sub-regions separated by a bottleneck can form an oscillator in this algorithm without convergence. In practice, we conjecture that this is a rare scenario and that our algorithm seems to reliably get "caught" in small convex sub-regions.

# B. Appendix: Details of Datasets and Extra Experimental Results

### B.1. Optimal Power Flow Datasets

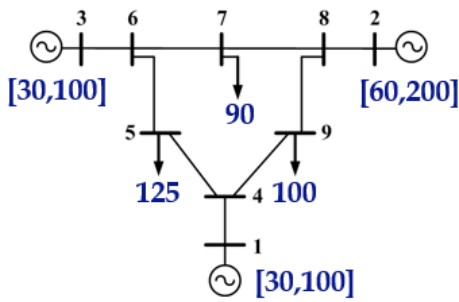

*Figure 10.* IEEE 9-Bus system topology . The system has 3 demand nodes (5,7,9) with nominal values (125, 90, 100) in MW, and the 3 generator nodes with (1,2,3) with limits ([30,100], [60,200], [30,100]) in MW. Therefore, the neural network has 3 inputs and 3 outputs.

We generate direct-current (DC) power flow datasets using the IEEE 9-bus power grid benchmark, a widely recognized model for simulating energy flow in power grids, depicted in Figure 10. Nodes 1, 2, and 3 are equipped with generators, while nodes 5, 7, and 9 serve as load points. Power operators can adjust the output at generator nodes, allowing for variations within predefined ranges. The neural network, $f_{\text{opf}}(P_D) = P_G$, predicts the electricity generated ($P_G$) at these three generator nodes based on the demand ($P_D$) from the load nodes and $P_D \in \mathbb{R}^3$. The dataset is created by solving the DC Optimal Power Flow (DC-OPF) problem (Frank et al., 2012) using nominal inputs $x_0 = [125, 90, 100]^T$ from standard datasets, perturbed by 10% uniform noise to produce a diverse set of 1,000 data samples. The DC-OPF specification requires that outputs stay within the physical limits of the generators, specifically $P_{G1} \in [30, 100]$, $P_{G2} \in [60, 200]$, and $P_{G3} \in [30, 100]$. We trained a three-layer ReLU neural network using the Adam optimizer (Kingma & Ba, 2014) to compute DC-OPF solutions, selecting 80% of the data samples randomly for training and reserving the remaining for testing.

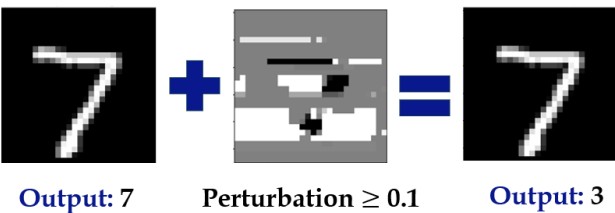

Output: 7          Perturbation ≥ 0.1          Output: 3

*Figure 11.* The minimum perturbation needed to the clean image of "7" (on the left) has a value of 0.1 and is shown in the center image. This results in an image (on the right) that is visually very close, but the classifier mistakes it for the digit "3".

## B.2. MNIST and CIFAR-10 Datasets

We utilize the MNIST digit dataset, which consists of 60,000 training and 10,000 testing samples, for an image recognition task (LeCun et al., 2010). MNIST is a widely used benchmark in neural network verification problems (Gowal et al., 2018; Wang & et.al., 2021; Kotha et al., 2024), comprising handwritten digits across ten classes (0 to 9). The neural network $f_{\text{image}}(x) = \hat{y}$ predicts class probabilities for each input $x$, and the goal is to ensure that the predicted probability for the true class $k$ exceeds that of any other class. This is operationalized as $\min_{k \neq l}(e_k - e_l)^T f_{\text{image}}(x)$.

We implement a two-layer ReLU neural network trained using cross-entropy loss and optimized with the Adam optimizer at a learning rate of 0.001. The network architecture for MNIST consists of three layers: input $x^0 \in \mathbb{R}^{784}$, two hidden layers $z^1, z^2 \in \mathbb{R}^{50}$, and output layer $z^3 \in \mathbb{R}^{10}$. The network achieves a test accuracy of 97%. For evaluation, we select the first image in the test dataset as the initial point $x^0$ for computing the maximum verifiable ball. Principal Component Analysis (PCA) is optionally used for MNIST to accelerate training, though it is not essential for the core results.

For the CIFAR-10 dataset (Krizhevsky et al., 2009), we use a subset of 10,000 test images. CIFAR-10 consists of $32 \times 32$ RGB images across ten object categories such as airplanes, cars, birds, and so on. Unlike MNIST, PCA is not used for dimensionality reduction due to the higher complexity and diversity of the image content.

Code and experiments are being made available at: https://github.com/LEVIS-LANL/LEVIS

## B.3. Visualization of the nearest adversarial point for image classification

For image recognition, we find the closest adversarial point to the clean image of the digit "7", in Figure 11. The distance to the adversarial point was 0.1. (in terms of pixel magnitude), and despite the image being visually the same, the classifier, even with its high accuracy of 93%, mistakes it for the digit "3".

## B.4. Statistical Performance of LEVIS-$\beta$

*Table 3.* Statistical Summary of the Radii

| Statistic | Value |
| --- | --- |
| Number of Radii | 52 |
| Minimum Radius | 0.10 |
| Maximum Radius | 23.58 |
| Median | 0.42 |
| Mean | 3.75 |
| Lower Quartile | 0.16 |
| Upper Quartile | 2.46 |
| Variance | 45.25 |
| Standard Deviation | 6.73 |

We provide more statistics on LEVIS-$\beta$ for DC-OPF. The main results are found in Table 3.

## B.5. CPU Resources

We used three machines for the computations. The specifications of the machines are shown in Table 4, Table 5, and Table 6

Table 4. CPU 1 Information Summary

| Attribute | Details |
| --- | --- |
| Processor | Intel(R) Core(TM) i7-8550U CPU @ 1.80GHz |
| Base Speed | 1.99 GHz |
| Current Speed | 1.57 GHz (can vary) |
| Cores | 4 |
| Logical Processors | 8 |
| L1 Cache | 256 KB |
| L2 Cache | 1.0 MB |
| L3 Cache | 8.0 MB |
| Virtualization | Disabled |
| Hyper-V Support | Yes |

Table 5. CPU 2 Information Summary

| Attribute | Details |
| --- | --- |
| Processor | Intel(R) Xeon(R) Gold 6258R CPU @ 2.70GHz |
| Base Speed | 2.70 GHz |
| Cores | 56 |
| Logical Processors | 112 |
| L1 Cache | 100 MB |
| L2 Cache | 3136 MB |
| L3 Cache | 154 MB |

Table 6. CPU 3 Information Summary

| Attribute | Details |
| --- | --- |
| Processor | Apple M3 Max CPU @ 64GHz |
| Cores | 16 |

