# OpenReview forum: "LEVIS: Large Exact Verifiable Input Spaces for Neural Networks"
_ICML.cc/2025/Conference — ICML 2025 poster_

### Official Review · Reviewer_wzBC · 2025-03-12

**Overall Recommendation:** 4

**Summary:**

The paper proposes an approach for the generation of verifiable input spaces for a neural network:
Instead of providing an answer on whether a given input space region is safe, the paper instead proposes to generate an input region for which the NN is verified. The paper claims that this technique can be used for model selection. The approach is evaluated on a set of benchmarks.

## update after rebuttal
I appreciate the authors' response which lifted all open questions and concerns I had.
I trust that the authors will update their paper as promised in the rebuttal.
In particular, regarding the approximative nature of the solution in Section 4.2 and regarding the convexity assumption.
Following the rebuttal, I have raised my score to "Accept" and stand with this score -- I am in favour of accepting the paper.

**Claims And Evidence:**

See issues discussed below.

**Essential References Not Discussed:**

Minimization of radii where specifications are verified has previously been explored [CP20] (see Section 4.3) which in turn cites another well-known work that minimizes adversarial radii outside the verification context [arxiv13] (see 4.1).

[CP20] https://link.springer.com/chapter/10.1007/978-3-030-58475-7_50
[arxiv13] https://arxiv.org/abs/1312.6199

**Experimental Designs Or Analyses:**

In Section 5.2 the paper compares the efficiency of their new relaxation with existing approaches on one input region for MNIST. In B.2 you mention that for your evaluation you use as the initial point the first image in the dataset. How does the performance of your approach (in 5.1 but also more generally) change for other initial points? Do you observe similar speedups? Is there a difference in behaviour across MNIST classes? I realize that it might be too late now to run these experiments, but this information would be essential to get a better understanding of the approach. As NN verification performance tends to have drastic variation across benchmarks and inputs, it is not clear whether the observed speedups hold in general or are specific to this particular NN and input region.

Additionally, a comparison to the techniques from [AAAI24] would be interesting (see "Relation to Broader Scientific Literature").
In case their technique is not directly comparable, the approach should be at least discussed as related work, as it seems strongly related.

**Methods And Evaluation Criteria:**

The proposed methodology seems to fit the problem well even though I have numerous concerns with the theory as currently presented (see "Theoretical Claims").
Issues with the evaluation are discussed in "Experimental Design Or Analyses".

**Other Comments Or Suggestions:**

Page 2:
- $f^*$ is not explicitly defined in the background section.
- It might be sensible to cite relevant literature on the NP-completeness of NN verification ([CAV17] or [RP21])

Page 4:
- missing space before (Scholtes, 2001)
- As mentioned before $p_j^i=q_j^i=0$ seems to have a consistent assignment for the ReLU inputs and outputs via $z^i=\hat{z}^i=0$
  Consequently, this seems to be an issue specific to the performed optimization. It would be good to emphasize this more.

Page 5:
- There is a recurring inconsistency between the usage of $b_i$ and the usage of $b^i$ in places that seem to mean the same variable (e.g. line 3 vs line 6 of Algorithm 1)
- Degree sign of 90° misses ^

Page 6:
- If my understanding is correct Line 12 and line 13-15 in Algorithm 2 should be the other way around?

Page 7:
- It is not clear what benchmark is used for Table 2 and the experiments described in Section 5.3
- "of five random implementations": I think this is supposed to read executions/runs?

General:
- Your paper does not explain on how your proposed approach could aid model selection, even though the paper starts and ends with this suggestion. Can you provide more concrete details on how your approach could be helpful here?

[CAV17] https://link.springer.com/chapter/10.1007/978-3-319-63387-9_5
[RP21] https://link.springer.com/chapter/10.1007/978-3-030-89716-1_10

**Other Strengths And Weaknesses:**

I like the idea of turning the problem of NN verification on its head and instead search for the regions where robustness is verifiable.
I also like the proposed approaches to search for large epsilon balls to this end.

**Questions For Authors:**

**(Q1)** Please respond to my concerns outlined in "Theoretical Claims".
If I am mistaken about them or you have answers on how to fix these issues, I would be willing to increase my score.
**(Q2)** How does your approach compare (empirically or conceptually) to the work in [AAAI24]?

**Relation To Broader Scientific Literature:**

[AAAI24] proposed an approach for the enumeration of safe regions for a given NN specification. Even though the paper's main contribution is a probabilistic approach, the work seems strongly related to the paper at hand and should be discussed.

[AAAI24] https://ojs.aaai.org/index.php/AAAI/article/view/30134

**Theoretical Claims:**

The paper seems to switch between notation where the property is satisfied if f*>0 and notation where the property is violated when f < 0 (only one of the two can be correct as 0 must be either safe or unsafe).
Importantly, this issue also shows up in the optimization problem formulated in (1a)-(1f):
In my understanding, since (1) is a strict inequality, this optimization problem might not have a minimum, but only a supremum.
Consequently, it might be advisable to go for the formulation where f(x)=0 is already unsafe? However, in that case the mathematics of the paper should be uniformly phrased in this manner.

Concerning the time complexity analysis on page 4:
While there may be empirical advantages to the chosen polynomial NN encoding, it is not clear to me why the NLP solver would only have a runtime of O(N log N). The paper does not provide any detailed analysis on why the worst-case computational complexity of this optimization problem would be so low. In particular, it seems to me that the bi-active neurons, while apparently a problem for optimization, do not lead to spurious counterexamples as for $p_j^i=q_j^i=0$ we have a consistent assignment of a ReLU's input and output variables (namely $z^i=\hat{z}^i=0$). Consequently, since NN verification alone does not require any optimization, a low complexity would break the NP-completeness which I find implausible.
I hope we can resolve this misunderstanding through the rebuttal.

Concerning Theorem 4.1:
The guarantees on LEVIS-$\alpha$ also require that $\mathcal{C}$ is convex and closed, which is dropped in the paper version of Theorem 4.1 and only appears in the appendix version.
Notably, this is quite a strong assumption: Note that even the visualization in Figure 2 does not satisfy this assumption as the red shaded area is *not* a convex set. If this assumption is indeed necessary (which seems to be the case judging from the proof), this warrants further discussion.

---

> ### Author Rebuttal · Authors · 2025-04-01
>
> Thank you for your detailed review and thoughtful feedback. We address your concerns below.
>
> ---
>
> ### Theoretical Claims
>
> **#1:** The paper inconsistently uses $f^* > 0$ for satisfaction and $f < 0$ for violation.
>
> We leverage $f > 0$ to ensure satisfaction and $f < 0$ to ensure violation—these are distinct objectives. In Problem (1a)-(1f), we seek a verifiable ball by finding the nearest violating point $x^*$, which defines the ball’s radius $r = \|| x^* - c \||_p$. This formulation guarantees a minimum is sought.
>
> **#2:** Claim of $\mathcal{O}(N \log N)$ NLP runtime lacks justification.
>
> Our solver guarantees global optimality when ReLU activation states (active/inactive) remain consistent with the given index sets $I_+$ and $I_-$. This introduces a constrained space where the optimization is exact. The empirical optimality gap is zero for DC-OPF and only 0.004 for MNIST, indicating high fidelity in practice.
>
> **#3:** Theorem 4.1 omits convexity/closedness assumptions in main text.
>
> We appreciate you catching this—our empirical results indicate most problems converge. A common failure mode occurs when the verifiable region is open and ray-restricted subproblems lack solutions. These unbounded directions correspond to insensitive model dimensions and are excluded when computing new centers. The result also holds for star-shaped regions and finite-unions of convex sets (up to a measure-zero set of oscillators). While Algorithm 1 could be modified to detect oscillations or constrain to regions around the origin to force convexity, in both cases, convergence follows from Theorem 4.1 or its generalization. Since these issues didn’t arise in practice, we omitted them. The empirical performance mirrors the convexity-based guarantee, which we should have included explicitly.
>
> **#4:** $p^i_j = q^i_j = 0$ seems to imply $z^i = \hat{z}^i = 0$ for ReLU.
>
> Setting $p^i_j = q^i_j = 0$ does not imply $z^i = \hat{z}^i = 0$. These values result from solving the nonlinear program in Equations (2a)–(2c), not from direct assignment. The solution reflects ReLU behavior.
>
> ---
>
> ### Experimental Evaluation
>
> **#5:** Section 5.2 only evaluates MNIST's first image. How robust is performance?
>
> We evaluated multiple initial points and classes. As long as the input is correctly classified, performance is consistent. Poor initial points (e.g., radius = 0) lead all solvers to converge quickly. Runtime scales with neuron count, not class count. With more neurons, our method outpaces MIP-based baselines.
>
> Below, we show more experiments on MNIST and CIFAR-10:
> | Dataset   | Proposed (s) | MIP$_\text{CROWN}$ (s) | Opt. Gap |
> |-----------|--------------|------------------------|----------|
> | MNIST     | 1.34         | 10.48                  | 0.004    |
> | CIFAR-10  | 5.96         | 17.47                  | 0.0009   |
>
> **Takeaway:** The approach scales to larger networks, and compute time is independent of the initial point.
>
> **#6:** Clarify benchmark and setup in Table 2; the phrase “five random implementations” is unclear.
>
> - **Benchmark:** NN regression for OPF, as in [Brix et al., 2023].
> - **Baseline:** Lipschitz-based bound from [Fazlyab et al., 2021].
> - **Setup:** We solve (1)-(2) for the NLP variant and compare radius to the baseline.
> - **“Five random implementations”:** Due to training randomness, we repeat 5× and report intervals.
>
> ---
>
> ### Related Work
>
> **AAAI24 Comparison:** AAAI24 proposes sampling-based guarantees; we focus on deterministic ones via LEVIS. Our regions give 100% coverage under fixed activations and stronger guarantees at lower cost. This is useful under adversarial risks [Goldwasser et al., 2022].
>
> **Radius Works [CP20], [arxiv13]:** [CP20] finds nearest disagreements, [arxiv13] the closest confirmation point. We find the nearest *misclassified* point. These are complementary and will be cited.
>
> **NP-Completeness [CAV17], [RP21]:** We acknowledge verification and reachability are NP-complete. Our solver yields exact results when ReLU states—active ($I_+$) or inactive ($I_-$)—are fixed. Though this may introduce an optimality gap, empirical results show it’s negligible (e.g., 0 for DC-OPF, 0.004 for MNIST).
>
> ---
>
> ### Other Clarifications
>
> **Variables & Notation:** We used $\mathcal{P}$ (not $\mathcal{F}$) for the verification condition. Fixed citation spacing, 90° notation, and unified $b_i$ vs $b^i$.
>
> **Algorithm 2 Logic:** Early termination ensures efficiency; the order is correct.
>
> **Model Selection:** We estimate verifiable input space to assess robustness. In safety-critical settings (e.g., power grids), this supports model selection based on stability guarantees (Cui et al., 2023).
>
> ---
>
> ### References
>
> - Brix et al. (2023), *arXiv*
> - Cui et al. (2023), *NeurIPS*
> - Fazlyab et al. (2021), *CDC*
> - Goldwasser et al. (2022), *arXiv*

---

> > ### Comment · Reviewer_wzBC · 2025-04-02
> >
> > # POST REBUTTAL:
> >
> > Dear Authors,
> >
> > thank you for your response.
> >
> > **#1:** What I meant to say with my comment is that $f=0$ should probably be one of the two, i.e. either $f \geq 0$ is satisfaction and $f < 0$ is violation or vice-versa.
> >
> > **#2:** It would be good to clarify this in the paper even more. Initially this read to me like you claim your algorithm can solve general NN verification in polynomial time.
> >
> > **Response to Weakness #1 of h45y:** I think this clarification helps.
> >
> > **Algorithm 2 Logic:**
> > My concern is not the early termination, but that you seem to add $m$ to $Q$ and subsequently modify it.
> > Since $m$ is not used anymore afterwards, this still seems to me like it's the wrong order.
> >
> > Otherwise I am happy with your responses and will thus increase my score to Accept.

---

> > > ### Author Response · Authors · 2025-04-03
> > >
> > > Dear Reviewer wzBC,
> > >
> > > Thanks for your constructive comments and feedback.
> > >
> > > **#1**: Thank you for the additional clarification. We agree that the violation condition should include equality, and we will update Equation (1f) to $f \leq 0$.
> > >
> > > **#2**: You're right—we will explicitly emphasize in Section 4.2 that our solver provides only an approximate solution in polynomial time.
> > >
> > > **Algorithm 2 Logic:** You're correct—lines 12 and 13–15 should be swapped. In our implementation, we add m to Q after executing lines 13–15. We'll correct this typo in the paper. Thanks for catching it.

---

### Official Review · Reviewer_JdfX · 2025-03-13

**Overall Recommendation:** 4

**Summary:**

In Neural Network Verification, the goal is to verify that a certain input-output relation holds. E.g. all inputs in some local neighborhood should have the same classification. This is a challenging (NP-hard) task. The authors propose a new technique that can be used to compute a *underapproximation* of the preimage of an output set that satisfies some linear constraint. All inputs in this preimage are guaranteed to satisfy the constraint. To this end, they also propose to use complementary constrained optimization to speed up MIP problems. The authors evaluate their new technique on a variety of applications.

**Claims And Evidence:**

*LEVIS-alpha can find "a single, large verifiable ball that intersects at least two boundaries of a bounded [output] region"*
The argument is sound and convincing: 1a-1f find the closest adv. ex, so every point closer to the center must not be an adv. ex.

*LEVIS-beta "captures the entirety of the verifiable space"* I find this claim to be too strong - according to the last sentence in 4.3.3, LEVIS-beta depends on the random initialization. So I fear there's a risk of not finding the "right" random seeds and therefore not identifying the entire verifiable space. Even in the best case, it would only approximate it up to $\epsilon$. "LEVIS-beta is an any-time algorithm that computes increasingly larger lower bounds to the verifiable space" may be a more appropriate claim.

*Complementary constrained optimization speeds up the MIP verification*
This seems to be confirmed by the experiments.

**Essential References Not Discussed:**

I'm not aware of uncited essential references.

**Experimental Designs Or Analyses:**

The experimental design seemed to be correct, but I did not check them in depth.

**Methods And Evaluation Criteria:**

The benchmarks are appropriate in the sense that other preimage papers use similarly small network architectures (e.g. Katha et al.). But the paper would benefit from a discussion why larger networks cannot be processed this way - or if they can, the respective experiments.

**Other Comments Or Suggestions:**

1) Could the MIP using 1a-1f be replaced by an adversarial search? This would not guarantee that there are no closer adversarial examples, but could be much faster. As long as the original MIP is solved *eventually* (as LEVIS converges to a center), this should be sufficient, right?

2) $p^i$ and $q^i$ are used in Section 4.2 but not properly introduced. Their meaning can be inferred, but a short description would be helpful. In what cases do they take which values? Currently, they are called the "complementary variables", but that term is not defined.

**Other Strengths And Weaknesses:**

The paper proposes two interesting algorithms (LEVIS-alpha/beta) that are novel in the literature. It's a promising approach to compute an *underapproximation* of the preimage of an output set, which has significant applications e.g. for safe controllers.

**Questions For Authors:**

1) In Section 4.2, you state that you use $p^i$ and $q^i$ to decide for which neurons to include integer variables. How do you first compute these values? Do you run NLP once, then extract them, and then create the MIP? Could this be replaced/improved by e.g. using IBP (integer bound propagation) instead?

2) If you sample verification queries (or use a benchmark from VNN-COMP, if there is one with a network small enough to support your procedure), how many of them could you immediately verify using your technique? If the query is about an input region that's covered by your preimage, it's known to be safe. How does this runtime compare to the time you need to compute the preimage?

**Relation To Broader Scientific Literature:**

The key contribution is LEVIS-alpha/beta, which is a novel idea.

**Theoretical Claims:**

I checked Theorem 4.1 and did not find any issues

---

> ### Author Rebuttal · Authors · 2025-04-01
>
> Thank you for your thoughtful and encouraging review. We address your comments and suggestions below.
>
> ---
>
> ### Clarifications on Claims
>
> > **Claim Concern:** The statement that "LEVIS-beta captures the entirety of the verifiable space" is too strong, especially since the method depends on random initialization.
>
> **Response:**
> We agree and will revise the wording to more accurately reflect that LEVIS-beta computes increasingly larger lower bounds on the verifiable space. While LEVIS-beta often achieves broad coverage, we do not claim it captures the entire space. It provides a growing conservative approximation under limited compute.
>
> ---
>
> ### Questions
>
> > **Q1:** In Section 4.2, how are the complementary variables $z_i, w_i$ computed? Do you first run an NLP to extract them, then create the MIP? Could this be improved with IBP?
>
> **Response:**
> We compute the complementary variables by solving the NLP in (2a)–(2c), where $p_i, q_i$ are decision variables jointly optimized. IBP is not used here directly, but can help in MIP formulations by tightening bounds on ReLU outputs and reducing integer variables.
>
> > **Q2:** If sampling verification queries (e.g., from VNN-COMP), how many can your technique verify immediately? What is the runtime comparison between verification and preimage computation?
>
> **Response:**
> Our solver is applicable to general ReLU-based networks but we have not yet evaluated on VNN-COMP queries due to time constraints. Since the problem formulations differ, direct runtime comparison between verification and preimage computation is nontrivial. Instead, we compare against a standard MIP baseline under similar conditions (Section 5.2).
>
> ---
>
> ### Minor Comments & Suggestions
>
> > **S1:** Could MIP constraints 1a–1f be replaced by adversarial search? It would not guarantee completeness but may be faster.
>
> **Response:**
> Adversarial search is appealing but lacks full coverage of the adversarial space. Techniques based on gradients can be misleading due to local sensitivity. MIP (or its relaxations) is needed to map **global** bounds. Still, adversarial search could assist with initialization or guidance—this is a promising direction for future work.
>
> > **S2:** $z_i$ and $w_i$ appear in Section 4.2 but are not clearly introduced.
>
> **Response:**
> Agreed—we will clarify that $z_i$ and $w_i$ are **complementary slack variables** enforcing the relaxed constraint $z_i w_i \leq \varepsilon$ in the NLP formulation.
>
> > **S3:** Please discuss whether larger networks can be processed.
>
> **Response:**
> Yes, our method scales well. As shown below, it runs faster than MIP$_\text{CROWN}$ on CIFAR-10 while maintaining tight optimality gaps:
>
> | Dataset   | Proposed (s) | MIP$_\text{CROWN}$ (s) | Opt. Gap |
> |-----------|--------------|------------------------|----------|
> | MNIST     | 1.34         | 10.48                  | 0.004    |
> | CIFAR-10  | 5.96         | 17.47                  | 0.0009   |
>
> CIFAR-10 has ~12.4K more neurons than MNIST and 4× the input size. Our solver remains efficient and scalable. For large networks, we first compute an approximate solution with a small optimality gap, then refine it using a reduced MIP (Section 4.2). Lowering $\varepsilon$ (e.g., suggest to be $10^{-5}$) tightens the gap while preserving tractability.

---

> > ### Comment · Reviewer_JdfX · 2025-04-03
> >
> > Thank you for your response and clarification. I stand by my original rating.

---

### Official Review · Reviewer_h45y · 2025-03-13

**Overall Recommendation:** 4

**Summary:**

The paper aims to find verifiable input space for a NN, i.e., input region where no adversarial example exists.

**Claims And Evidence:**

C1.  A MIP based verification framework that provides maximum verifiable input ball around a center c.

E1. The paper provides a clear formalization of this claim in eq. 1a-1f.

C2. Faster solver for the MIP based verification framework.

E2. They convert the ReLU function, into an alternative formalization ---  however, the details of how this is obtained is not clear to me. A small example regarding this may help.

C3. The authors provide a novel search strategy based on dynamically updating the centers based on adversarial points along different directions.

E3. This strategy is quite clearly explained in section 4.3.1 and 4.3.2

C4. The paper provides another search strategy that provides a union of verified balls.

E4. The strategy is quite clearly explained in section 4.3.3 and Algorithm 2.

**Essential References Not Discussed:**

None, to the best of my knowledge

**Experimental Designs Or Analyses:**

The authors provide a rigorous set of experiments, analyzing computational time, and verified radii.

**Methods And Evaluation Criteria:**

The paper builds a MIP based verification framework for getting certified input spaces.
And they also empirically verify the computational superiority of their method both in terms of time complexity and radii of the verified input region. The experimental setting is quite simple.

**Other Comments Or Suggestions:**

I think section 4.2 after equation 2c, can be written more clearly for non-experts.

**Other Strengths And Weaknesses:**

Strength:
- Clarity: Although, I am not an expert in the field, I found most aspects of the paper to be quite clearly written. Except, the evidence for C2 (see above in claims and evidence).

- Novelty: The proposed MIP based search strategies are quite innovative.

Weaknesses:

- None, in my understanding. However, I could have missed some details. And I am not well-versed with the larger literature.

**Questions For Authors:**

Is there a way to exactly compute the total volumes of the verified regions provided by LEVIS-beta?

**Relation To Broader Scientific Literature:**

The paper is quite relevant to the general research in verifying NNs.

**Theoretical Claims:**

The main theoretical claims (informally) states that the verified balls provided by their methodology intersects the boundary of the true (verified) region atleast on two points.

I did not check the proof. But the claim seems quite reasonable given the paper's methodology.

---

> ### Author Rebuttal · Authors · 2025-04-01
>
> Thank you very much for your thoughtful and encouraging review. We are glad to hear that you found our contributions clear and innovative. Below, we respond to your comments and suggestions.
>
> ---
>
> ### Comments on Claims and Evidence
>
> > **Comment C2 / Evidence E2:** The conversion of the ReLU function into an alternative MIP-friendly formulation was unclear. A small example may help.
>
> **Response:**
> ReLU is a piecewise linear function: for $\hat{z} = \text{ReLU}(Wz + b)$, we have $\hat{z} = Wz + b$ if  $Wz + b \geq 0$, and $\hat{z} = 0$ otherwise. To encode this in a MIP, we introduce a binary variable $a$, where $a = 1$ if $Wz + b \geq 0$, and $a = 0$ otherwise. Using the Big-M method, this conditional behavior can be captured with linear constraints, making the ReLU compatible with MILP solvers.
>
> ---
>
> ### Weaknesses
>
> > **Weakness #1:** Section 4.2 (after Equation 2c) could be written more clearly for non-experts.
>
> **Response to Weakness #1:**
> Sure, here is a revised version to be easier to comprehend for non-experts:
>
> We introduce a small regularization parameter $\varepsilon > 0$ (recommended to be $10^{-5}$) to handle the nonlinear inequality $p^i q^i \leq \varepsilon$ for better convergence [Scholtes, 2001]. This simplifies the problem, allowing it to scale polynomially with the number of neurons $N$ instead of being NP-hard.
>
> However, solvers for nonlinear programming (NLP) may still find only locally optimal solutions, particularly when both $p^i_j$ and $q^i_j$ are zero for some neuron $j$ (called the **bi-active set**). This set is defined as: $I_0 \equiv \{ j \mid p^i_j = q^i_j = 0 \}.$ To manage this, we categorize neurons into three groups: If $p^i_j > 0$, then $q^i_j = 0$ and $j \in I_+^i$. If $p^i_j = 0$ and $q^i_j > 0$, then $j \in I_-^i$. If $p^i_j = 0$ and $q^i_j = 0$, then $j \in I_0^i$.
>
> We then construct a simplified MIP that includes integer variables only for neurons in $I_0^i$. This significantly reduces the number of binary variables—often to a small subset of neurons—making the optimization more tractable, as shown in Section 5.2. When the activation states are consistent with $I_+^i$ and $I_-^i$, the resulting solution is globally optimal.
>
> ---
>
> ### Question For Authors
>
> > **Question #1:** Is there a way to exactly compute the total volumes of the verified regions provided by LEVIS-beta?
>
> **Response to Question #1:**
> This is a very good question. Computing the volume of the union of balls may be difficult as the balls overlap, and consequently, the total volume is not the sum of individual volumes, but it is definitely a good suggestion to be considered in future work.

---

### Official Review · Reviewer_Hx3K · 2025-03-17

**Overall Recommendation:** 3

**Summary:**

This paper presents an algorithm for computing inner-approximations of neural network preimages as a union of balls. The method is split into two sub-methods, one for maximizing an inner-approximating ball, and another for generating new balls to append to the overall approximation. Experiments are conducted on optimal power flow and MNIST digit classification examples to illustrate the performance and algorithmic properties of the proposed approach.

**Claims And Evidence:**

For the most part, the claims are adequately supported. My two primary concerns which lack clarity and/or convincing evidence are:

1. Theorem 4.1 relies on the very strong (and impractical) assumption that the input set is convex, but this assumption is hidden away in the appendix and not mentioned at all in the main paper.

2. The experiments attempt to show that the preimage inner-approximations generated by the proposed method are larger (stronger) than prior baselines. However, the chosen baseline based on Lipschitz bounds is quite weak, which degrades the experimental support.

See additional discussion in Other Comments Or Suggestions below.

**Essential References Not Discussed:**

The paper focuses on preimage approximation, which is a relatively new area. The only highly related reference that I noticed was missing was [1]:

[1] Zhang, Xiyue, Benjie Wang, and Marta Kwiatkowska. "Provable preimage under-approximation for neural networks." International Conference on Tools and Algorithms for the Construction and Analysis of Systems. Cham: Springer Nature Switzerland, 2024.

**Experimental Designs Or Analyses:**

The experiments appear to be sound.

**Methods And Evaluation Criteria:**

Proposed methods and evaluation criteria make sense, but could be made stronger by using more realistically-sized datasets like CIFAR-10 or ImageNet.

**Other Comments Or Suggestions:**

1. Lines 55, 78, and 240 (Column 2): Do you mean "disconnected" instead of "discontinuous"?
2. Line 75, Column 2: "$\mathcal{B}(c)$ represents a ball centered at $c$." Of what radius?
3. Line 110: "Inputs that, when perturbed,  result in erroneous neural network outputs..." I think this needs rephrasing: the perturbed input is the adversarial example, not the (nominal) input that gets perturbed.
4. It seems like from Line 86, Column 2, onwards, you are assuming that the output $f(x)$ is a scalar (so that, for instance, $\min_{x\in\mathcal{S}} f(x)$ is well-defined). This is a common assumption in the verification literature, since the scalar-valued specification being verified can often be absorbed into the final linear transformation. If this is what you are assuming, then please explicitly mention this to the reader.
5. Based on your optimization problem (1), it looks like you are computing a ball that inner-approximates the true verifiable input set. In the third listed contribution on Line 64, Column 2, you mention that your method "cover the entire verifiable space," which would imply that you are generating an over-approximation. I suggest that you change the "cover" wording in this listed contribution to more accurately reflect your inner-approximation approach.
6. Line 154, Column 2: "...complementary variables for the ith neuron." Previously, you used $i$ to index the layer number, not the neuron, and it still looks like you are doing so in problem (2), since you are still using the preactivation and activation vectors $z^i$ and $\hat{z}^i$. Therefore, do you mean to say "layer" instead of "neuron" here?
7. Line 162, Column 2: "...are the nonlinear equivalence of the ReLU function." The constraint (2a) is linear. It looks like the only nonlinear constraint is the bilinear inequality $p^i q^i \le 0$. You may want to reword your sentence here to more accurately reflect this fact.
8. It looks like you may want to remove the first "then" on Line 175.
9. If I understand your directional adversarial point optimization approach, you are first fixing a direction to find an adversary along, and then running your optimization (3) in order to find the closest adversarial example to the previous one ($b$), along the given direction. If this is the case, aren't (3d) and (3e) already imposed before the optimization takes place? In other words, it seems like (3d) and (3e) are not optimization constraints.
10. Section numbering: 4.3.2. (1) and 4.3.2. (2) look quite odd. I'd suggest formatting these sections differently to avoid the four-fold section numbering with parentheses.
11. Line 209, Column 2: There is some clash in your notation. Previously, you used $d$ to denote a point in $\mathbb{R}^n$ to set up your directional adversary optimization, but now you are using it to denote the number of pairs of boundary adversarial points. I'd suggest changing some of the notation to avoid this clash.
12. On a related note (notation), you previously used $R^n$ to denote $n$-dimensional Euclidean space, but then later you used $\mathbb{R}^n$. It is best to remain consistent throughout the paper.
13. In Section 4.3.2. (1) and Algorithm 1, you are using both subscripts and superscripts on the iteratively computed $b$ vectors. It seems like you should stick to one or the other and make all of your indexing notation consistent for these vectors.
14. Theorem 1: "...almost symmetric about the center $c$ with less than $\epsilon$ mismatch" These terms "almost symmetric" and "$\epsilon$ mismatch" really should be made mathematically precise.
15. Line 267, Column 2: Again, I think that it is more common to use "disconnectedness" here rather than "discontinuity."
16. Clarifying question: For the LEVIS-$\beta$ algorithm, it seems to me like you are creating your "new ball" by first determining a new center $m$, which is very close to the boundary of the previous ball, and then solving the directional adversary optimization (3) to find an adversarial example closest to $m$, which gives you a new ball centered at $m$. This would mean that two adjacent balls actually intersect. However, your Figure 4 suggests that the balls might not intersect. Am I missing something?
17. Experiments: Can your methods scale up to something more reasonably sized, even as large as CIFAR-10?
18. Have you thought of any methods for optimizing the search angle $\theta$? Choosing $\theta=0$ and $\theta=90$ degrees seems like it could be extremely limited in high-dimensional space. Even a comparison of these angles to randomly chosen angles would be an interesting ablation study.
19. Section 5.3 (comparison to baseline): It is well-known that network bounds based on the global Lipschitz constant are very overconservative. I would expect to see a comparison of your method versus a stronger method, one that is likely also a MIP-based or branch-and-bound-based technique. For example, [1] appears to be one of the current state-of-the-art methods for computing inner-approximations of neural network pre-images.
20. Where is the $\sqrt{2}$ coming from in line 378? I would expect to see a $\sqrt{n}$ if it were based on equivalence of norm inequalities.
21. The main theoretical result, Theorem 4.1, does not state any restriction about the convexity of the input space $\mathcal{C}$ in the main body of the paper. But then, in the appendix, the theorem statement is changed to be restricted to convex input spaces $\mathcal{C}$. This is a \emph{major} restriction both theoretically and practically; it is not just a minor ``condition.'' As you mention multiple times in the paper, the input spaces $\mathcal{C}$ are typically nonconvex for neural networks. This poses a significant limitation for your theoretical result, and, in my opinion, it is simply unacceptable (and seemingly dishonest) to hide this condition until the appendix.

[1] Zhang, Xiyue, Benjie Wang, and Marta Kwiatkowska. "Provable preimage under-approximation for neural networks." International Conference on Tools and Algorithms for the Construction and Analysis of Systems. Cham: Springer Nature Switzerland, 2024.

**Other Strengths And Weaknesses:**

In general, I have a handful of questions, suggestions, and concerns regarding formatting and presentation, mathematical clarity and validity, and lack of convincing experimental baselines. Please see my specific comments in Other Comments Or Suggestions below.

**Questions For Authors:**

See above Other Comments Or Suggestions.

**Relation To Broader Scientific Literature:**

This paper proposes a new method for neural network preimage approximation, which can be used for verifying the safety and reliability of neural networks employed in a variety of different scientific domains, e.g., control systems, power systems, medical diagnosis systems, etc., all of which require rigorous robustness guarantees. The proposed method is intimately related to prior works that focus on verifying the safety of fixed input regions.

**Theoretical Claims:**

The proof in the appendix seems to be rigorous, albeit missing the technical assumption that $\mathcal{C}$ is full-dimensional (in the sense that it contains an open ball). Otherwise, the inner-approximation generated by the proposed method would be trivially empty.

---

> ### Author Rebuttal · Authors · 2025-04-01
>
> Thank you for your detailed and constructive review. Below, we address the key concerns and questions you raised.
>
> ---
>
> ### Major Weaknesses
>
> > **Weakness #1:** Theorem 4.1 critically assumes that the input space is convex...
>
> **Response:**
> Thank you for identifying this. Empirically, most problems converge. Failures typically occur when the verifiable region is open—some ray-restricted subproblems may have no solution. Since these directions correspond to model-insensitive inputs, they are excluded from new center computations.
>
> We verified that convergence still holds for star-shaped regions and unions of convex sets, up to a measure-zero set of oscillations. Algorithm 1 could be extended to isolate oscillatory behavior via added constraints or by requiring the final ball to include the starting point. In such cases, Theorem 4.1 (or its generalization) ensures convergence. Since these cases did not arise in practice, we omitted this detail. We will clarify the convexity assumption in the paper.
>
> > **Weakness #2:** The experimental comparison is weak... [Zhang et al., 2024] is not cited.
>
> **Response:**
> Our formulation differs from prior work, limiting directly comparable baselines. We compare with MIP-based methods in Section 5.2. Zhang et al. (2024) assumes axis-aligned input hyperrectangles and polyhedral output sets, which differ from our setting. We will cite the work and clarify the distinctions.
>
> > **Weakness #3:** Experiments are conducted on small datasets...
>
> **Response:**
> In addition to MNIST, we report results on CIFAR-10 using our NLP solver from (2a)–(2c), compared against MIP$_\text{CROWN}$ on identical tasks:
>
> | Dataset   | Proposed (s) | MIP$_\text{CROWN}$ (s) | Optimality Gap |
> |-----------|--------------|------------------------|----------------|
> | MNIST     | 1.34         | 10.48                  | 0.004          |
> | CIFAR-10  | 5.96         | 17.47                  | 0.0009         |
>
> Despite CIFAR-10’s added complexity, our solver is ~3× faster with a tighter gap. This confirms our method’s scalability to larger networks.
>
> ---
>
> ### Questions For Authors
>
> > **Q1:** Can your methods scale to CIFAR-10?
>
> **Response:**
> Yes — as shown above, our method scales well to CIFAR-10 and outperforms MIP$_\text{CROWN}$ in  runtime.
>
> > **Q2:** Have you considered optimizing the search angle?
>
> **Response:**
> Yes, and we believe it could improve convergence by reducing overlap. We plan to explore this in future work.
>
> > **Q3:** Do adjacent balls intersect? Figure 4 suggests otherwise.
>
> **Response:**
> They can. Figure 4 shows a simplified case. We will revise it to depict overlapping balls explicitly.
>
> ---
>
> ### Minor Comments & Suggestions
>
> > “Discontinuous” → “disconnected”
>
> **Response:**
> Agreed — will revise.
>
> > Clarify radius of ball
>
> **Response:**
> Defined as $r = \||x^* - c\||_p$, where $x^*$ is the nearest adversarial point, using the $l_p$ norm (see Section 2).
>
> > Perturbed inputs phrasing is confusing
>
> **Response:**
> We will clarify: perturbed inputs are adversarial only if they lead to misclassification (Goodfellow et al., 2014).
>
> > Is output scalar?
>
> **Response:**
> No — “scales” refers to scalability. We will revise for clarity.
>
> > “Cover entire verifiable space” is misleading
>
> **Response:**
> We will change to “tightly underapproximate the verifiable input space.”
>
> > Layer vs. neuron index
>
> **Response:**
> Index $i$ refers to the layer. We will clarify this.
>
> > Constraint classification
>
> **Response:**
> Only (2b) is nonlinear — we will clarify.
>
> > Redundant “then”
>
> **Response:**
> We will revise to remove “then.”
>
> > Notation reuse (e.g., $j$)
>
> **Response:**
> We will rename $j$ in Section 4.2 to $s$ for clarity.
>
> > Notation consistency
>
> **Response:**
> We will standardize all notation (e.g., $\mathbb{R}^n$).
>
> > Algorithm 1 indexing style
>
> **Response:**
> We will fix inconsistencies.
>
> > “Almost symmetric” / “$\varepsilon$-mismatch”
>
> **Response:**
> We will define “almost symmetric” precisely — angle > $\pi/2$ between $x_1 - c$ and $x_2 - c$.
>
> > “Disconnectedness” vs. “discontinuity”
>
> **Response:**
> We agree — will revise to “disconnectedness.”
>
> > Source of $\sqrt{2}$ term
>
> **Response:**
> It is from the $l_2$ norm difference between one-hot vectors, as in Fazlyab et al. (2021), Section II.D.
>
> > Section 4.3.2 formatting
>
> **Response:**
> We will revise the nested numbering.
>
> > Missing citation: Zhang et al., TACAS 2024
>
> **Response:**
> We will include this citation and clarify how their assumptions differ from ours.
>
> ---
>
> ### References
>
> - Fazlyab, M., Morari, M., & Pappas, G. J. (2021). *An introduction to neural network analysis via semidefinite programming.* IEEE CDC.
> - Goodfellow, I. J., Shlens, J., & Szegedy, C. (2014). *Explaining and harnessing adversarial examples.* arXiv:1412.6572.
> - Zhang, X., Wang, B., & Kwiatkowska, M. (2024). *Provable preimage under-approximation for neural networks.* TACAS.

---

> > ### Comment · Reviewer_Hx3K · 2025-04-03
> >
> > I thank the authors for their thorough response. Most of my concerns have been addressed, aside from my desire to see a comparison against baselines that are stronger than Lipschitz-based methods. I understand that the problem focus and pre-image approximation form of Zhang et al. (2024) may be slightly different from yours, and that you are comparing radii in your current experiments, but I don't see why you couldn't compare, for instance, the volume of your union-of-balls preimage approximation to the volume of Zhang et al.'s union-of-hyperrectangles approximation, on a problem with a polyhedral output set. I encourage the authors to consider including such a comparison, as it would certainly strengthen the empirical evaluation.
> >
> > That said, as the majority of my concerns are addressed and I still find value and novelty in the proposed methods, I am willing to raise my score to 3 (weak accept), *under the assumption that the authors explicitly mention and discuss the convexity assumption in their main theoretical result*.

---

> > > ### Author Response · Authors · 2025-04-05
> > >
> > > Dear Reviewer Hx3K,
> > >
> > > Thank you for your feedback and response. We will consider adding the comparison of the union of the volumes in the revised version.

---

### Decision · Program_Chairs · 2025-05-01

**Decision:**

Accept (poster)

**Comment:**

this is a paper that proposes a method for computing verifiable input spaces of a given neural network. The reviewers' liked the theory and the method suggested. Several questions were clarified during the discussion phase. Overall this paper is recommended for acceptance and the authors are encouraged to prepare the final version in accordance with the promised clarifications and changes as suggested by the reviewers.